# Zfhx4 regulates endochondral ossification as the transcriptional platform of Osterix in mice

Eriko Nakamura [1✉], Kenji Hata [1], Yoshifumi Takahata [1], Hiroshi Kurosaka[2], Makoto Abe[3], Takaya Abe [4], Miho Kihara[4], Toshihisa Komori[5], Sachi Kobayashi[1], Tomohiko Murakami [1], Toshihiro Inubushi[2], Takashi Yamashiro[2], Shiori Yamamoto[1], Haruhiko Akiyama[6], Makoto Kawaguchi[7], Nobuo Sakata [8] & Riko Nishimura [1✉]

Endochondral ossification is regulated by transcription factors that include SRY-box transcription factor 9, runt-related protein 2 (Runx2), and Osterix. However, the sequential and harmonious regulation of the multiple steps of endochondral ossification is unclear. This study identified zinc finger homeodomain 4 (Zfhx4) as a crucial transcriptional partner of Osterix. We found that *Zfhx4* was highly expressed in cartilage and that *Zfhx4* deficient mice had reduced expression of matrix metallopeptidase 13 and inhibited calcification of cartilage matrices. These phenotypes were very similar to impaired chondrogenesis in *Osterix* deficient mice. Coimmunoprecipitation and immunofluorescence indicated a physical interaction between Zfhx4 and Osterix. Notably, *Zfhx4* and *Osterix* double mutant mice showed more severe phenotype than *Zfhx4* deficient mice. Additionally, Zfhx4 interacted with Runx2 that functions upstream of Osterix. Our findings suggest that Zfhx4 coordinates the transcriptional network of Osterix and, consequently, endochondral ossification.

[1] Department of Molecular and Cellular Biochemistry, Osaka University Graduate School of Dentistry, 1-8 Yamadaoka, Suita, Osaka 565-0871, Japan. [2] Department of Orthodontics and Dentofacial Orthopedics, Osaka University Graduate School of Dentistry, 1-8 Yamadaoka, Suita, Osaka 565-0871, Japan. [3] Department of Oral Anatomy and Developmental Biology, Osaka University, Graduate School of Dentistry, 1-8 Yamadaoka, Suita, Osaka 565-0871, Japan. [4] Laboratory for Animal Resources and Genetic Engineering, RIKEN Center for Biosystems Dynamics Research, 2-2-3 Minatojima Minami-machi, Chuou-ku, Kobe 650-0047, Japan. [5] Basic and Translational Research Center for Hard Tissue Disease, Nagasaki University Graduate School of Biomedical Sciences, Nagasaki 852-8588, Japan. [6] Department of Orthopaedic Surgery, Gifu University Graduate School of Medicine, Yanagido 1-1, Gifu 501-1194, Japan. [7] Division of Diagnostic Pathology, Niigata Rosai Hospital, Japan Organization of Occupational Health and Safety, 1-7-12 Toh-un-cho, Johetsu, Niigata 942-8502, Japan. [8] Department of Biochemistry, Showa Pharmaceutical University, Machida, Tokyo 194-8543, Japan. ✉email: enakamura@dent.osaka-u.ac.jp; rikonisi@dent.osaka-u.ac.jp

Vertebrate skeletal elements are formed through two distinct processes: intramembranous and endochondral ossifications[1,2]. Intramembranous ossification is the process by which mesenchymal cells differentiate into osteoblasts and is consequently responsible for the formation of the clavicle and most of the craniofacial skeleton[3]. Conversely, endochondral ossification involves numerous steps of chondrocyte differentiation followed by replacement of cartilage tissue with bone tissue, which gives rise to most bones that include vertebrae, ribs, and long bones[1,4]. The first step of endochondral ossification is mesenchymal condensation after which cells differentiate into resting and proliferating chondrocytes that produce components of the chondrogenic extracellular matrix including collagen type 2 alpha 1 chain (*Col2a1*), collagen type 11 alpha 2 chain (*Col11a2*), and aggrecan (*Acan*)[5,6]. The chondrocytes then differentiate into hypertrophic chondrocytes that produce collagen type 10 alpha 1 chain (*Col10a1*), Indian hedgehog protein (*Ihh*), and matrix metalloproteinase 13 (*Mmp13*)[7–10]. Subsequently, hypertrophic chondrocytes undergo apoptosis, cartilage matrices are calcified, vascular vessels invade the cartilage tissue and, finally, cartilage tissues are replaced by bone tissue[11]. These unique and complex steps are regulated by various critical transcription factors and coactivators[10].

The transcription factor SRY-box transcription factor 9 (Sox9), is essential for the early stage of chondrogenesis, in which mesenchymal condensation and chondrocyte differentiation occur[12]. Mutations in the *SOX9* gene in humans cause campomelic dysplasia (OMIM 114290) characterized by severe chondrodysplasia[13,14]. Similarly, mutations around the *SOX9* gene cause Pierre Robin sequence (OMIM 261800) that involves a cleft palate[15]. The expression of transcriptional factors Sox5 and Sox6, both of which are crucial for early chondrogenesis, is induced by Sox9[16–18]. A critical role of Sox9 in chondrogenesis has also been demonstrated by a study in which chondrogenesis was severely impaired in chondrocyte-specific *Sox9* conditional knockout (KO) mice[19]. Moreover, Sox9 directly regulates the expression of early-stage chondrogenic genes that include *Col2a1*, *Col11a2*, and *Acan*[6,16,20].

Conversely, runt-related protein (Runx) 2, Runx3, and Osterix are crucial for the late stages of chondrogenesis. Chondrocyte hypertrophy is severely impaired in *Runx2* KO mice, whereas *Runx2* and *Runx3* double KO mice exhibit a complete lack of hypertrophic chondrocytes[21]. Furthermore, Runx2 plays a critical role in the regulation of *Col10a1*, *Ihh*, and vascular endothelial growth factor (*Vegf*)[21–23]. Osterix (also known as Sp7) functions as a downstream transcriptional partner of Runx2 during bone and cartilage development[10]. Chondrocyte-specific *Osterix* conditional KO mice show a lack of calcification in cartilage matrices and matrix vesicles as well as loss of *Mmp13* expression[10]. Thus, the Sox9-Runx2-Osterix axis is responsible for spatiotemporal mediation of endochondral ossification.

Transcriptional partners of Sox9 and Runx2 have been identified, which interact with the proteins during endochondral ossification. Peroxisome proliferator-activated receptor coactivator 1α forms a complex with Sox9[24] similarly to p300/CREB-binding protein[25], protein 54 kDa nuclear RNA-binding protein (p54$^{nrb}$)[26], AT-rich interactive domain (Arid) 5a[27], Arid5b[28], zinc finger protein 219[29], and WW domain-containing E3 ubiquitin protein ligase 2 (Wwp2)[30,31]. Core-binding factor beta (Cbfβ) forms a heterodimer with Runx2[21,32,33], which also interacts with CCAAT/enhancer binding protein β[34].

Considering the sequential and harmonious regulation of endochondral ossification achieved through this transcriptional network, it is possible that transcription factors that have not been identified may be involved in the multiple steps of cartilage development.

The present study aimed to identify novel transcription factors involved in the regulation of endochondral ossification and to elucidate the functional roles of these factors. We isolated zinc finger homeobox 4 (Zfhx4) from mouse limb buds and identified it as a highly expressed transcription factor in chondrogenic tissues. We demonstrated that Zfhx4 is required for endochondral ossification in conjunction with Osterix. We also found that Zfhx4 physically interacts with Runx2. Because Zfhx4 is a large transcription factor with four homeodomains and 22 zinc finger domains, Zfhx4 might function in the transcriptional platform and interact with several transcription factors and transcriptional regulators. Our findings provide insights into the role of Zfhx4 in the regulation of endochondral ossification.

## Results

**Identification of Zfhx4 as a transcription factor highly expressed in chondrocytes.** Microarray analysis revealed high expression levels of *p54$^{nrb}$*, *Arid5b*, *Bbf2h7*, *Cbfβ*, *ATF4*, *Sox9*, *Sox5*, *Sox6*, *Wwp2*, *Runx2*, *Dlx5*, *Dlx6*, *Msx2*, and *Osterix* in mouse limb bud cells, all of which play important roles in endochondral ossification (Fig. 1a). Furthermore, *Zfhx4* was highly expressed in limb bud cells (Fig. 1a). We focused on Zfhx4 for further analyses because of its putative role in 8q21.11 microdeletion syndrome (OMIM 614230) characterized by skeletal abnormalities[35]. Figure 1b shows the results of reverse transcriptase-quantitative polymerase chain reaction (RT-qPCR) analyses of newborn mouse tissues, which showed that *Zfhx4* was highly expressed in several skeletal tissues that included ribs, limbs, and the calvaria. Consistent with the RT-qPCR data, expression of *Zfhx4* was detected in limb bud cells by whole mount in situ and in situ hybridization analyses (Supplementary Fig. 1a, b). Additionally, in situ hybridization showed high expression of *Zfhx4* in the growth plate of embryonic day (E) 16.5 mouse femurs (Fig. 1c).

**Presence of skeletal abnormalities and cleft palate in Zfhx4-deficient mice.** To generate *Zfhx4*-deficient mice, we first generated *Zfhx4* floxed mice (Supplementary Fig. 1c) and confirmed germline transmission by Southern blot and PCR analyses (Supplementary Fig. 1d). The generation of *Zfhx4* heterozygous-deficient mice by further crossing was confirmed by PCR.

*Zfhx4* homozygous deficient (*Zfhx4$^{-/-}$*) mice died within 1 day after birth, exhibited a cleft palate (Fig. 1d), and became cyanotic with gasping respirations and air-distended stomachs (Supplementary Fig. 2). Moreover, *Zfhx4$^{-/-}$* mice exhibited skull malformations and shorter humeri and femurs than wild-type littermates and suffered loss of the condylar process and hypoplasia of the tracheal–bronchial ring at P0 (Supplementary Figs. 3a and 4). *Zfhx4$^{-/-}$* embryos showed skeletal hypoplasia of the skull, mandible, humerus, femur, and ribs at E15.5 and E16.5 (Fig. 1e and Supplementary Fig. 3b). Notably, delayed ossification of the humerus and femur of *Zfhx4$^{-/-}$* mice was observed at E15.5 and E16.5 (Fig. 1e and Supplementary Fig. 3b).

**Impaired endochondral ossification in Zfhx4-deficient mice.** Histological analysis revealed that E15.5 *Zfhx4$^{-/-}$* mice had very few hypertrophic chondrocytes, although wild-type littermates exhibited appropriate hypertrophic chondrocyte zones (Fig. 2a). Additionally, E16.5 *Zfhx$^{-/-}$* mice showed impairment of calcification in cartilage matrices (Fig. 2b). In situ hybridization revealed normal expression of *Sox9* and *Col2a1* in E15.5 *Zfhx4$^{-/-}$* mice. However, such expression was not separated at the diaphysis (Fig. 2c). *Ihh*, *Ppr*, *Runx2*, and *Osterix* were expressed, but not separated at the diaphysis in *Zfhx4$^{-/-}$* mice compared with wild-type littermates (Fig. 2c). Most strikingly, no expression of

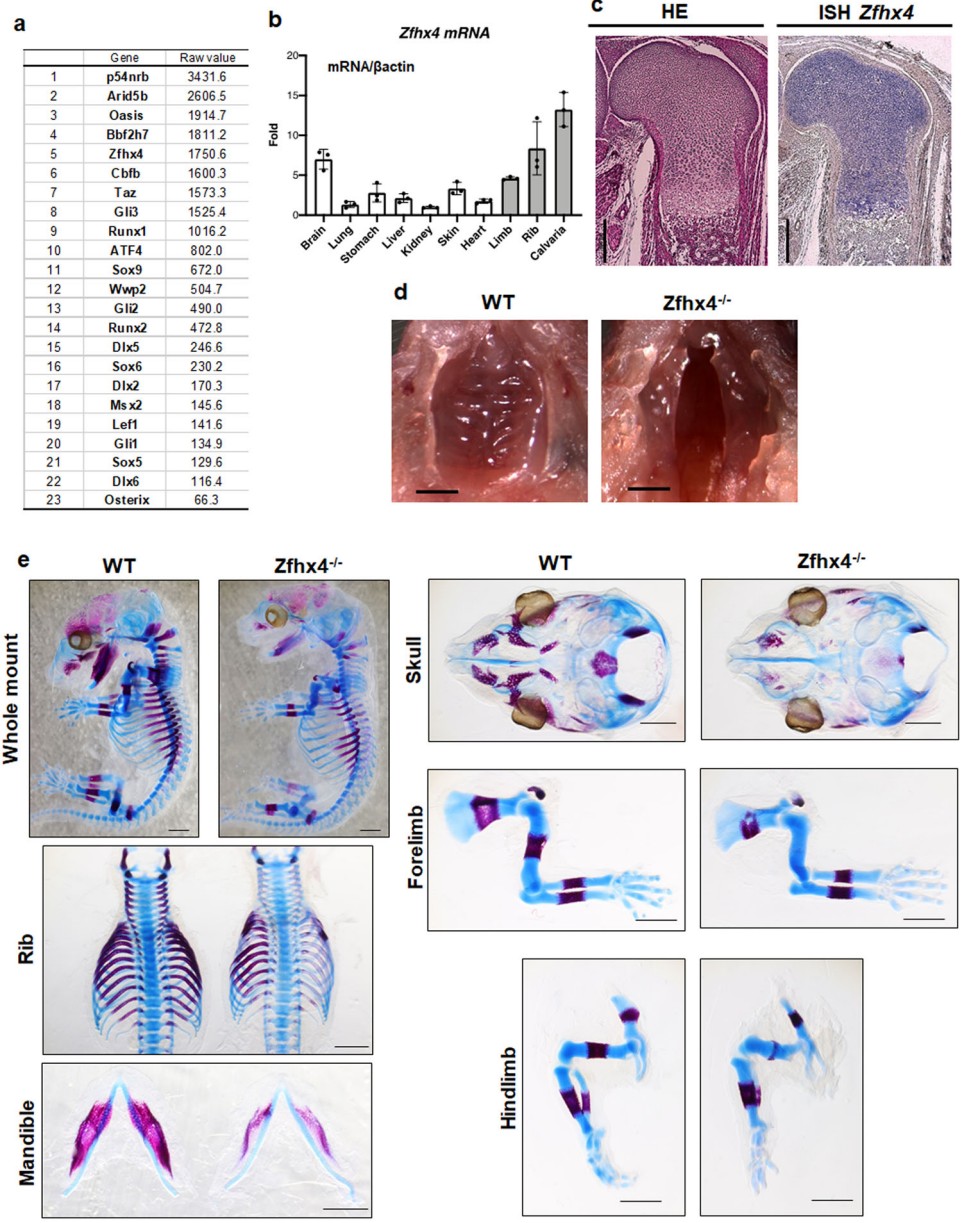

**Fig. 1 Identification of Zfhx4 as an important transcription factor for skeletal development. a** List of highly expressed transcription factors in limb buds determined by microarray analysis. **b** Expression of *Zfhx4* in tissues of newborn mice. Total RNA was isolated from the indicated tissues and analyzed by RT-qPCR. Data are shown as the mean ± s.d. (*n* = 3). **c** Expression of *Zfhx4* in the growth plate of mice. Hematoxylin and eosin staining (HE) and in situ hybridization (*ISH*) were carried out on growth plate chondrocytes of femurs from E16.5 mice. Scale bar = 500 μm. **d** Macroscopic views of the palate of P0 wild-type (WT) and *Zfhx4*$^{-/-}$ littermates. Cleft palates were observed in *Zfhx4*$^{-/-}$ mice. Scale bar = 1 mm. **e** Skeletal preparations of whole embryos, ribs, mandibles, skulls, forelimbs, and hindlimbs of E15.5 WT and *Zfhx4*$^{-/-}$ littermates stained with Alcian blue and alizarin red. Scale bar = 1 mm.

*Col10a1* or *Mmp13* was detected in *Zfhx4*$^{-/-}$ mice (Fig. 2c). Immunofluorescence staining and RT-qPCR analyses also showed very little expression of Mmp13 in *Zfhx*$^{-/-}$ mice (Fig. 2d, e).

**Interaction of Zfhx4 with Osterix during endochondral ossification.** Because the skeletal phenotype of *Zfhx4*$^{-/-}$ mice resembled that of *Osterix* KO mice with reduced *Mmp13* expression and calcification of cartilage matrices[10], we hypothesized that Zfhx4 interacted with Osterix during endochondral ossification. Coimmunoprecipation and immunofluorescence analyses demonstrated that Zfhx4 physically associated with Osterix and that the proteins were colocalized in the nucleus (Fig. 3a, b). These results clearly demonstrated an association of Zfhx4 with Osterix. To understand the importance of this finding

in terms of endochondral ossification, we generated double mutant mice of *Zfhx4* and *Osterix*. Because *Osterix* KO mice exhibit a complete lack of *Mmp13* expression and calcification of cartilage matrices[10], we attempted to generate *Zfhx4*$^{-/-}$; *Osterix*$^{+/-}$ mutant mice and compared the phenotype with *Zfhx4*$^{-/-}$ mice. Double mutant *Zfhx4*$^{+/-}$;*Osterix*$^{+/-}$ mice appeared to be similar to wild-type mice (Fig. 3c–f, Supplementary Fig. 5a). Conversely, *Zfhx4*$^{-/-}$;*Osterix*$^{+/-}$ mutant mice had less calcification of the humerus and femur at E16.5 than *Zfhx4*$^{-/-}$ littermates (Fig. 3c). Histological analyses indicated that hypertrophy and calcification of the femur were more severely impaired in *Zfhx4*$^{-/-}$;*Osterix*$^{+/-}$ mice at E16.5 than in *Zfhx4*$^{-/-}$ littermates (Fig. 3d, e, Supplementary Fig. 6). Moreover, Mmp13 expression was diminished in *Zfhx4*$^{-/-}$;*Osterix*$^{+/-}$ mice even at

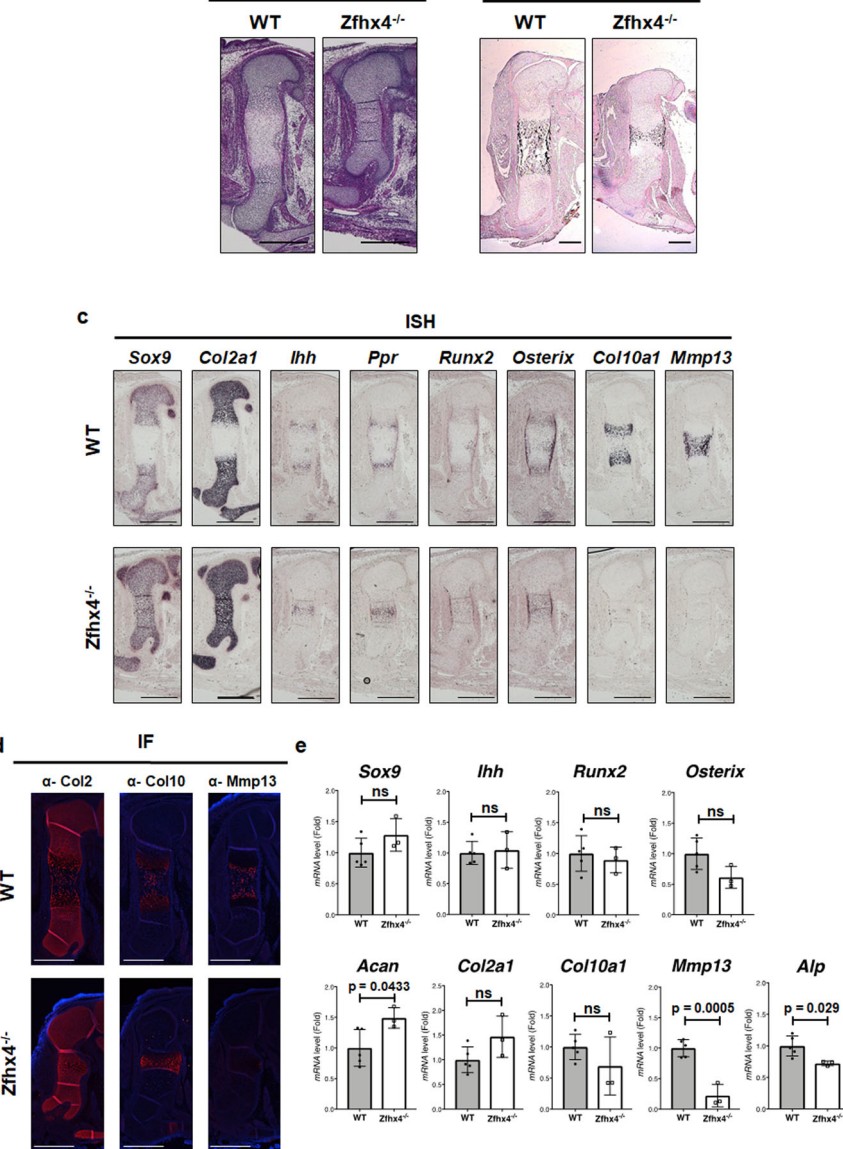

**Fig. 2 Delayed endochondral ossification and downregulation of Mmp13 expression in Zfhx4-deficient mice. a** Hemolysin and eosin (HE)-stained femurs of E15.5 wild-type (WT) and *Zfhx4$^{-/-}$* littermates. Scale bar = 500 µm. **b** Von Kossa-stained femurs of E16.5 WT and *Zfhx4$^{-/-}$* littermates. Scale bar = 200 µm. **c** Expression of chondrogenic gene markers in *Zfhx4$^{-/-}$* mice. Paraffin-embedded sections of femurs from E15.5 WT and *Zfhx4$^{-/-}$* littermates were examined by in situ hybridization using anti-sense probes against *Sox9, Col2a1, Ihh, Ppr, Runx2, Osterix, Col10a1,* and *Mmp13*. Scale bar = 500 µm. **d** Immunofluorescence analysis of femurs from E15.5 WT and *Zfhx4$^{-/-}$* littermates using anti-Col2, anti-Col10, and anti-Mmp13 antibodies. Scale bar = 500 µm. **e** mRNA expression levels of chondrocyte marker genes. mRNA isolated from humeri of E15.5 WT and *Zfhx4$^{-/-}$* littermates was analyzed by RT-qPCR. Data are presented as the mean ± s.d. (WT: n = 5; KO: n = 3).

E16.5 when *Zfhx4$^{-/-}$* littermates exhibited marginal expression of Mmp13 (Fig. 3f). *Zfhx4$^{-/-}$;Osterix$^{+/-}$* mice also showed more severe impairment at E15.5 than *Zfhx4$^{-/-}$* littermates (Supplementary Fig. 5b, c).

It has been shown that *Mmp13* expression is regulated by a complex formed by Runx2 and Osterix[10]. Supplementary Figure 7a shows the results of coimmunoprecipitation of Zfhx4 and Runx2, which revealed that the two proteins interacted physically and were localized to the nucleus (Supplementary Fig. 7b). Consistent the notion that Runx2 is an upstream transcription factor of Osterix[10,36], Osterix was not required for the interaction of Zfhx4 with Runx2 (Supplementary Fig. 7c, d). We next examined the role of Zfhx4 in regulation of the *Mmp13* gene promoter. A luciferase reporter assay indicated that Zfhx4

upregulated *Mmp13* gene promoter activity and increased transcriptional activities of Osterix and Runx2 on their promoters (Supplementary Fig. 8a). Consistently, chromatin immunoprecipitation analyses revealed binding of Zfhx4, Osterix, and Runx2 to *Mmp13* gene promoter regions (Supplementary Fig. 8b). Because Runx2 had a different binding ability to the *Mmp13* gene promoter from Zfhx4 and Osterix, Zfhx4 may sequentially interact with Runx2 and Osterix.

**Role of Zfhx4 in palate development.** Because *Zfhx4$^{-/-}$* mice showed a complete cleft palate at 100% penetrance (Fig. 1d, Supplementary Fig. 9a), we examined expression of *Zfhx4* in the palatal shelf. In situ hybridization analysis clearly indicated that *Zfhx4* was expressed in the palatal shelf of E13.5 mice (Fig. 4a).

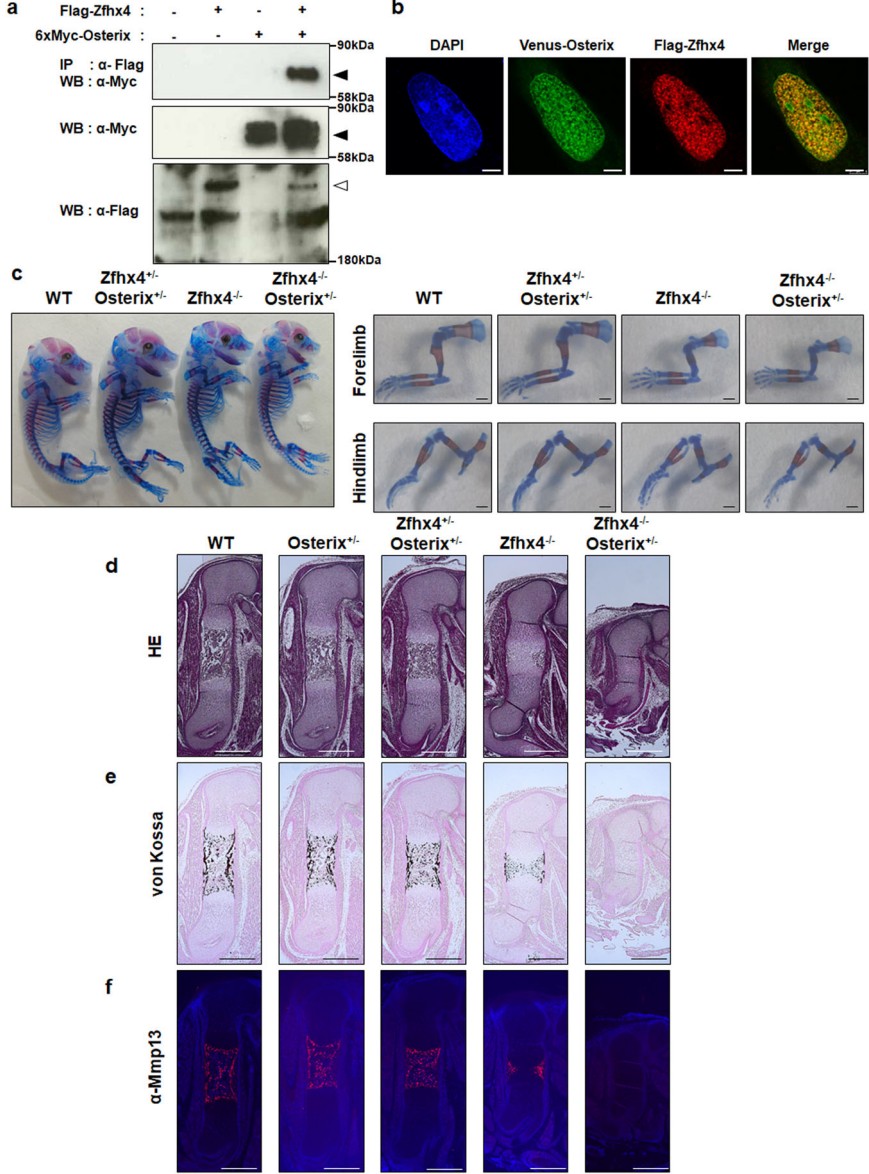

**Fig. 3 Importance of the interaction between Zfhx4 and Osterix for late-stage endochondral ossification. a** Physical interaction of Zfhx4 with Osterix. Coimmunoprecipitation analysis of lysates of 293FT cells transfected with Flag-Zfhx4, 6xMyc-Osterix, or both is shown. Closed arrows: 6xMyc-Osterix, Open arrow: Flag-Zfhx4 (IP immunoprecipitation, WB western blotting). **b** Colocalization of Flag-Zfhx4 (red) with Venus-Osterix(green) in SW1353 cells. Scale bar = 5 μm. **c** Skeletal preparations of whole bodies, forelimbs, and hindlimbs of E16.5 wild-type (WT), $Zfhx4^{+/-};Osterix^{+/-}$, $Zfhx4^{-/-}$, and $Zfhx4^{-/-};Osterix^{+/-}$ littermates stained with Alcian blue and alizarin red. Scale bar = 1 mm. Right panels show higher magnification images of the forelimbs and hindlimbs shown in the left panel. **d–f** Paraffin-embedded sections of E16.5 WT, $Osterix^{+/-}$, $Zfhx4^{+/-};Osterix^{+/-}$, $Zfhx4^{-/-}$, and $Zfhx4^{-/-};Osterix^{+/-}$ littermates subjected to hematoxylin and eosin (HE) staining (**d**), von Kossa staining (**e**), and immunofluorescence staining with an anti-Mmp13 antibody (**f**). Scale bar = 500 μm.

Examination of palate development of wild-type and $Zfhx4^{-/-}$ littermates at E13.5, E14.5, and E16.5 (Fig. 4b and Supplementary Fig. 9b–d) revealed that the growth and elongation of palatal shelves were normal in both $Zfhx4^{-/-}$ and wild-type E13.5 mice. Palatal shelves were fused in E14.5 wild-type mice, whereas palatal shelves of $Zfhx4^{-/-}$ mice were not elevated and had failed to fuse by E14.5 (Fig. 4b and Supplementary Fig. 9c). Palatal shelves of E16.5 $Zfhx4^{-/-}$ mice were degraded (Fig. 4b and Supplementary Fig. 9d). Consistent with the results indicating that growth and elongation of palatal shelves were normal in $Zfhx4^{-/-}$ mice, immunofluorescence with anti-Pcna and -Ki67 antibodies indicated normal cell proliferation in the palatal shelf of $Zfhx4^{-/-}$ mice (Fig. 4c, d). Additionally, there was no detectable difference in the number of apoptotic cells in the

palatal shelf between $Zfhx4^{-/-}$ and wild-type littermate mice (Supplementary Fig. 10), which suggested that apoptosis was not involved in the etiology of cleft palate in $Zfhx4^{-/-}$ mice. To further examine the role of Zfhx4 in palatal development, we performed organ culture experiments using maxilla of E14.2 $Zfhx4^{-/-}$ mice. Unexpectedly, palatal selves of $Zfhx4^{-/-}$ mice were elevated and fused as well as wild-type littermates (Supplementary Fig. 11). These results suggested that Zfhx4 was unnecessary to elevate the palatal shelf itself.

Palatal development involves multiple steps, which include palatal shelf outgrowth, elevation, adhesion, and fusion, and palatal bone formation. Odd-skipped-related (Osr) 1[37,38], Osr2[37,38], fibroblast growth factor 10 (Fgf10)[39], msh homeobox 1 (Msx1)[40,41], and Paired box 9 (Pax9)[42,43] play essential roles in

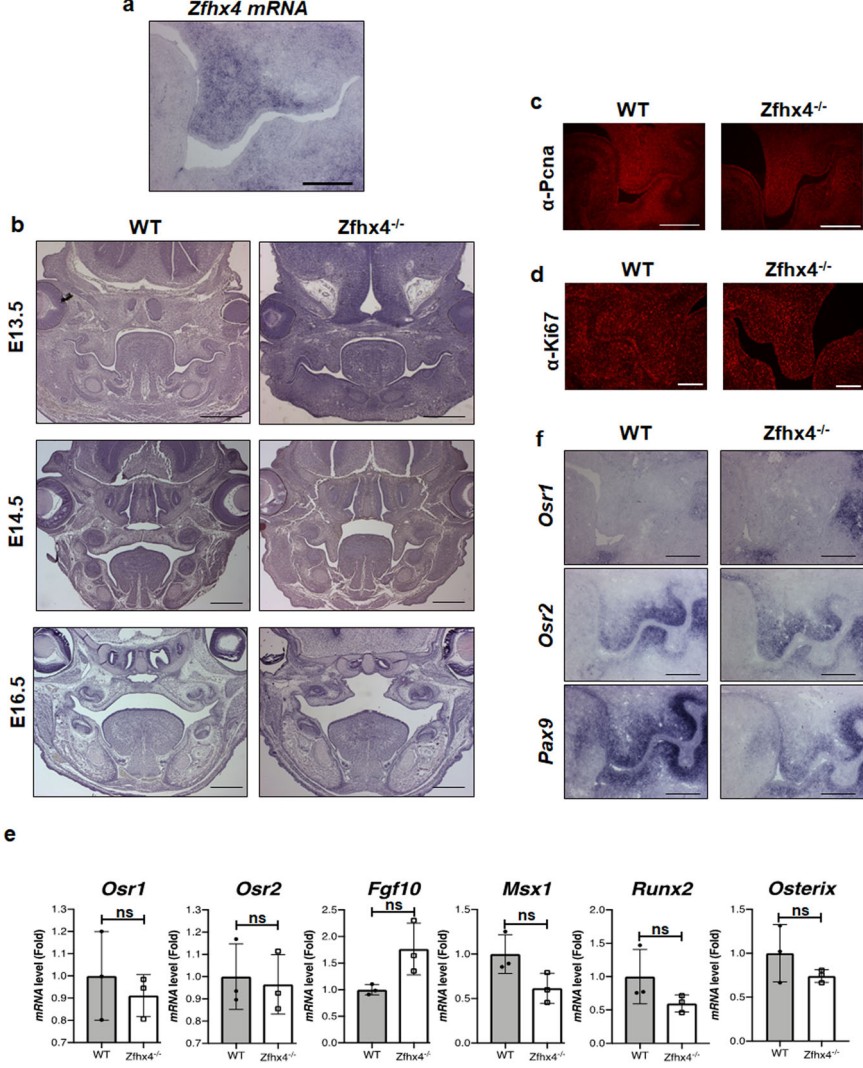

**Fig. 4 Critical role of Zfhx4 in palate development. a** *Zfhx4* expression in the palatal shelf of E13.5 mice determined by in situ hybridization. Scale bar = 200 μm. **b** Coronal sections of crania of E13.5, E14.5, and E16.5 wild-type (WT) and *Zfhx4*−/− littermates stained with hematoxylin and eosin (HE). Scale bar = 500 μm. **c** Fluorescence microscopy images of coronal sections of crania of E13.5 WT and *Zfhx4*−/− littermates subjected to immunofluorescence analysis using an anti-Pcna antibody. Scale bar = 100 μm. **d** Coronal sections of crania of E13.5 WT and *Zfhx4*−/− littermates subjected to immunofluorescence analysis using an anti-Ki67 antibody. Scale bar = 100 μm. **e** Expression levels of genes associated with palatal development in the palatal shelves of E13 WT and *Zfhx4*−/− littermates as determined by RT-qPCR using TaqMan probes. Data represent the mean ± s.d. (WT: *n* = 3; KO: *n* = 3). **f** Coronal sections of crania of E13.5 WT and *Zfhx4*−/− littermates were examined by in situ hybridization using anti-sense probes against *Osr1*, *Osr2*, and *Pax9*. Scale bar = 200 μm.

palatal development. Cranial neural crest-specific *Sox9* conditional KO mice[44] and *Runx2* KO mice[45] exhibit cleft palates. In the present study, we found no significant difference between wild-type and *Zfhx4*−/− littermates in terms of the expression of *Osr1*, *Osr2*, *Fgf10*, *Msx1*, or *Pax9* in palatal shelves (Fig. 4e, f and Supplementary Fig. 12). The expression of *Runx2* and *Osterix* also remained unchanged in *Zfhx4*−/− compared with wild-type littermates (Fig. 4e). Therefore, these results suggested that expression of these genes was not regulated and unknown Zfhx4 target genes might be involved in palatal development. Identification of target genes of Zfhx4 in the palatal shelf may contribute to understanding the molecular basis of palatal development, which involves Zfhx4.

## Discussion

Cell differentiation, proliferation, and death are strictly regulated by transcriptional machinery complexes that comprise specific transcriptional factors and transcriptional coregulators. Although transcriptional regulation during endochondral ossification has been extensively investigated[1], the details of the sequential and harmonious mechanism have not yet been elucidated. To address this, we attempted to identify transcription factors that regulate the transcriptional network system in endochondral ossification. In this study, we identified Zfhx4 as a transcriptional factor that is highly expressed in cartilage and limb buds. Notably, we found that Zfhx4 functioned as a transcriptional partner of Osterix. Coimmunoprecipitation and immunofluorescence analyses indicated a physical interaction between Zfhx4 and Osterix. Furthermore, genetic experiments revealed that Zfhx4 and Osterix interacted functionally in the late stage of endochondral ossification. Our data also indicated that *Mmp13* expression was regulated by Zfhx4. Additionally, luciferase reporter and chromatin immunoprecipitation analyses indicated regulation of the *Mmp13* gene promoter by Zfhx4. Because Osterix is critical for the regulation of *Mmp13* expression in chondrocytes[10], our

results support the importance of the association between Zfhx4 and Osterix in endochondral ossification.

*Col10a1* expression was diminished in *Zfhx4*[−/−] mice as determined by in situ hybridization experiments (Fig. 2c). Conversely, we observed similar expression level of *Col10a1* and Col10 between wild-type and *Zfhx4*[−/−] littermates (Fig. 2d and e), although the delayed expression pattern of Col10 was consistent with the in situ hybridization experiments. We assume that these difference resulted from slightly different time points of these experiments. Therefore, we believe that *Col10a1* is not a critical transcriptional target for Zfhx4.

Because of the structure of Zfhx4, it is possible that Zfhx4 associates with numerous transcriptional regulators that function in the platform of the transcriptional network during endochondral ossification. In addition to Osterix, we found that Zfhx4 physically interacted with Runx2, a critical transcription factor in endochondral ossification[46]. Furthermore, Runx2 functions upstream of Osterix by controlling *Mmp13* expression through its interaction with Osterix[10]. Thus, our study suggests that Zfhx4 facilitates the Runx2-Osterix axis, which results in coordination of the sequential steps of the transcriptional network that underlies endochondral ossification.

We found that endochondral ossification was impaired in the proximal limbs of *Zfhx4*-deficient mice, which included the scapula, humerus, and femur. The anterior–posterior patterning of the proximal limb is regulated by homeobox (Hox) 9 and 10 paralogous groups[47–50] as well as short stature homeobox 2 (Shox2)[51]. These genes are expressed in a spatiotemporally restricted pattern in domains along the axes of limb buds and provide anterior–posterior positional information for proximal limb patterning. *Hoxa9*/*Hoxd9* double mutant mice have shortened humeri and loss of deltoid crests[50], and triple mutant mice (*Hoxa10*/*Hoxc10*/*Hoxd10*) show short humeri, loss of deltoid processes, and remarkably shortened femurs[49]. Mutation of *Shox2* causes marked shortening of the humerus and femur, as well as loss of the deltoid crest and ossification of the femur[52]. Double mutation of *HoxD* and *Shox2* (*HoxD*[−/−];*Shox2*[+/−]) causes the phenotype of shortened humeri and lack of deltoid crests, whereas *HoxD*[+/+];*Shox2*[+/−] mutants show intact proximal limbs[51]. Considering the morphological phenotypes of *Hox9*, *Hox10*, and *Shox2* mutant mice, Zfhx4 may be involved in the patterning processes of proximal limbs, possibly in association with Hox9, Hox10, and Shox2. Clinically, shortening of limbs is classified into four types: rhizomelic (dysplasia of proximal parts of the limbs), mesomelic (dysplasia of middle parts of the limbs), acromelic (dysplasia of distal parts of the limbs), and micromelic (generalized shortening of entire limbs)[53]. On the basis of the skeletal phenotype of *Zfhx4*-deficient mice, this mutation induces rhizomelic type skeletal dysplasia. Rhizomelic type skeletal dysplasia, achondroplasia, rhizomelic chondrodysplasia punctata, and spondyloepiphyseal dysplasia congenita are well characterized and several genes have been identified to be responsible for these diseases[53]. Achondroplasia (OMIM100800) is caused by heterozygous mutations in the gene that encodes fibroblast growth factor receptor 3[54], rhizomelic chondrodysplasia punctata type1 (RCDP1) (OMIM 215100) is caused by homozygous or compound heterozygous mutations in the gene that encodes peroxisomal biogenesis factor 7 (*PEX7*)[55], and spondyloepiphyseal dysplasia congenital (OMIM183900) is caused by heterozygous mutations in *COL2A1*[56]. However, the causative genes and pathological mechanisms of some rhizomelic type skeletal dysplasias, which include cleidorhizomelic syndrome, Patterson-Lowry rhizomelic dysplasia, and multiple epiphyseal dysplasia with Ribbing type, are yet to be elucidated. The *Zfhx4* mutation described in the present study may be a good model to evaluate the pathogenesis of rhizomelic type skeletal dysplasia and such

assessment might provide insights into the pathogenesis of rhizomelic dysplasia.

Our finding that *Zfhx4* is highly expressed in calvariae suggests that Zfhx4 plays roles in both membranous and endochondral ossifications. However, *Zfhx4*-deficient mice showed no remarkable phenotype in terms of membranous ossification. We therefore assume that Zfhx4 has a partially redundant role and that its function might be compensated by other genes. A potential candidate gene is zinc finger homeobox 3 (Zfhx3; also known as ATBF1) that belongs to the Zfhx family and has the highest homology with Zfhx4 of all family members. The Zfhx3 protein is 406 kDa and contains four homeodomains and 23 zinc finger domains[57]. Supporting our theory, P10 *Zfhx3* heterozygous mice are much smaller in terms of body size and weight than wild-type littermates[58]. This is presumably due to impaired skeletal development. Although dissection of *Zfhx3*/*Zfhx4* double mutant mice may provide insights into the potential roles of Zfhx family proteins in skeletal development, this was beyond the scope of the present study. We intend to carry this out as the next step of this study.

We unexpectedly identified cleft palate at the elevation stage in *Zfhx4*-deficient mice. Notably, *Zfhx4* was specifically expressed in palatal shelves, but not in the tongue. However, organ culture experiments showed that elevation and fusion abilities of palatal shelves from *Zfhx4*[−/−] mice and wild-type littermates were intact. An explanation for this discrepancy between in vivo and ex vivo experiments is that Zfhx4 is required to remove the tongue from the middle of developing palatal shelved to create the space for palatal shelves to elevate and fuse. Considering specific expression of *Zfhx4* in palatal shelves, it is likely that Zfhx4 regulates the expression of genes involved in palatal development. However, the expression levels of well-known palatal marker genes appeared normal in *Zfhx4*-deficient mice. Further investigation is required to identify Zfhx4 target genes in palatal shelves. The *Zfhx4* gene has been proposed to be a candidate gene responsible for 8q21.11 microdeletion syndrome[35] and a report has shown this disease manifesting as a cleft palate in eight unrelated individuals. Thus, *Zfhx4*-deficient mice might be a suitable animal model to investigate the molecular mechanisms of palatal development and the pathogenesis of 8q21.11 microdeletion syndrome.

In conclusion, our results demonstrate that Zfhx4 is a transcriptional partner of Osterix and regulates the late stage of endochondral ossification. Our findings contribute to improving our understanding of the sequential and harmonious regulation of endochondral ossification.

## Methods

**Cell culture**. Anterior and posterior limb buds were dissected from E12.5 Institute of Cancer Research (ICR) mouse embryos (SLC, Shizuoka, Japan) and digested by incubation in Dulbecco's modified Eagle's medium (DMEM) (Sigma–Aldrich, St. Louis, MO, USA) with 0.1% collagenase type II (Sigma–Aldrich) and 0.1% trypsin (Sigma–Aldrich) at 37 °C[29]. Cells were dissociated by pipetting and then centrifuged at $300 \times g$ for 5 min. The supernatant was removed and the cell pellet was resuspended with α-minimum essential medium (Sigma–Aldrich) with 10% fetal bovine serum (FBS; JRH Bioscience, Lenexa, KA, USA) to a final density of $2 \times 10^7$ cells/ml. Cells were incubated at 37 °C in a humidified atmosphere with 5% $CO_2$. 293FT and SW1353 cell lines were purchased from Thermo Fisher Scientific (Waltham, MA, USA) and the American Type Culture Collection (Manassas, VA, USA), respectively. Cells were cultured in DMEM with 10% FBS at 37 °C in a humidified atmosphere with 5% $CO_2$.

**Animal experiments**. All animal experiments were approved by the Osaka University Graduate School of Dentistry animal care committee and the Institutional Animal Care and Use Committee (IACUC) of RIKEN Kobe Branch. Animal experiments with E12, E12.5, E13.5, E14.5, E15.5, and E16.5 embryos, and newborn littermates were performed by following protocols approved by the Osaka University Graduate School of Dentistry animal care committee. CAG-Cre transgenic

**Table 1 List of sequences of Taqman probe sets for real-time RT-PCR experiments.**

| *Zfhx4* | Sense | 5′-TGGCCCCACCAAATCTTTGC-3′ |
|---|---|---|
| | Anti-sense | 5′-TGGCTCGGTCTGCTGTCC-3′ |
| | Probe | 5′-TCCACCACCACCTCCTCCTCCTCCA-3′ |
| *Sox9* | Sense | 5′-CCTTCAACCTTCCTCACTACAGC-3′ |
| | Anti-sense | 5′-GGTGGAGTAGAGCCCTGAGC-3′ |
| | Probe | 5′-CCGCCCATCACCCGCTCGCAATAC-3′ |
| *Ihh* | Sense | 5′-GACTCATTGCCTCCCAGAACTG-3′ |
| | Anti-sense | 5′-CCAGGTAGTAGGGTCACATTGC-3′ |
| | Probe | 5′-CCACAGCCAGCCTGGACATCCCGA-3′ |
| *Runx2* | Sense | 5′-CTCCTTCCAGGATGGTCCCA-3′ |
| | Anti-sense | 5′-CTTCCGTCAGCGTCAACACC-3′ |
| | Probe | 5′-CACCACCTCGAATGGCAGCACGCT-3′ |
| *Osterix* | Sense | 5′-AGCGACCACTTGAGCAAACAT-3′ |
| | Anti-sense | 5′-GCGGCTGATTGGCTTCTTCT-3′ |
| | Probe | 5′-CCCGACGCTGCGACCCTCCC-3′ |
| *Aggrecan* | Sense | 5′-TCACTGTTACCGCCACTTTCC-3′ |
| | Anti-sense | 5′-TGCTGCTCAGATGTGACTGCC-3′ |
| | Probe | 5′-ACCGTCTCTCCGCATCCACCCAGG-3′ |
| *Col2a1* | Sense | 5′-CCTCCGTCTACTGTCCACTGAG-3′ |
| | Anti-sense | 5′-TGGAGCCCTGGATGAGCAAG-3′ |
| | Probe | 5′-TGAGGTTGCCAGCCGCTTCGTCCA-3′ |
| *Col10a1* | Sense | 5′-GCCAAGCAGTCATGCCTGAT-3′ |
| | Anti-sense | 5′-GACACGGGCATACCTGTTACC-3′ |
| | Probe | 5′-AGCACTGACAAGCGGCATCCCAGA-3′ |
| *Mmp13* | Sense | 5′-GGTTATGACATTCTGGAAGGTTATCC-3′ |
| | Anti-sense | 5′-CGTGGTTCTCAGAGAAGAAGAGG-3′ |
| | Probe | 5′-CCCGTGTTCTCAAAGTGAACCGCAGCG-3′ |
| *Alp* | Sense | 5′-ATCTTTGGTCTGGCTCCCATG-3′ |
| | Anti-sense | 5′-TTTCCCGTTCACCGTCCAC-3′ |
| | Probe | 5′-TGAGCGACACGGACAAGAAGCCCTT-3′ |
| *Osr1* | Sense | 5′-CCACCTACGGGACCACAGATATA-3′ |
| | Anti-sense | 5′-CGGGACTGGCAGAATCCTTT-3′ |
| | Probe | 5′-CCACACTCTTGACACTTGAAAGGCTTCTCT-3′ |
| *Osr2* | Sense | 5′-CACCTACGGGATCACAGGTATATC-3′ |
| | Anti-sense | 5′-GCAGCTGTAGGGCTTGATGT-3′ |
| | Probe | 5′-ACTGACAGAATCCTTTCCCACACTCCTGAC-3′ |
| *Fgf10* | Sense | 5′-CCTGGAGATAACATCAGTGGAAATC-3′ |
| | Anti-sense | 5′-TCCTCTATTCTCTCTTTCAGCTTACA-3′ |
| | Probe | 5′-CCGTCAAAGCCATCAACAGCAACTATTACT-3′ |
| *Msx1* | Sense | 5′-AAGATGCTCTGGTGAAGGCC-3′ |
| | Anti-sense | 5′-TCTTGTGCTTGCGTAGGGTG-3′ |
| | Probe | 5′-CGTGGATGCAGAGTCCCCGCTTCTC-3′ |
| *β-actin* | Sense | 5′-TTAATTTCTGAATGGCCCAGGTCT-3′ |
| | Anti-sense | 5′-ATTGGTCTCAAGTCAGTGTACAGG-3′ |
| | Probe | 5′-CCTGGCTGCCTCAACACCTCAACCC-3′ |

mice (RBRC01828) were provided by the RIKEN BioResource Research Center (Ibaraki, Japan).

*Zfhx4* mutant mice were designed with floxed exons 7 and 8 (Supplementary Fig. S1c). Deletion of exons 7 and 8 caused a nonsense frame-shift mutation in most functional domains of the Zfhx4 protein, including all homeobox domains, 13 zinc finger domains and a region coding nuclear localization signal. To generate *Zfhx4* floxed mice (accession number CDB0954K, http://www2.clst.riken.jp/arg/micelist.html), TT2 embryonic stem (ES) cells were transfected with a targeting vector that contained a neomycin resistance gene (Supplementary Fig. S1c) by electroporation[59]. Homologous recombination was confirmed by Southern blot analysis. The ES cells were injected into eight-cell-stage ICR embryos and *Zfhx4* floxed chimera mice were obtained. The *Zfhx4* floxed chimeras were mated with C57BL/6 mice and then germline transmission of the mutant allele was achieved as confirmed by Southern blotting and genomic PCR analyses. The 590 bp probe (Supplementary information) located in exon 10 of the *Zfhx4* gene (Supplementary Fig. 1c) was used for Southern blotting analyses of EcoRI-digested genomic DNA from mouse tails. The sequences of primers used for genomic PCR were as follows: 5′-GCCAAAGGCTGACTCAAAAC-3′ and 5′-GGGTCCCCACTGTGATTTCT-3′. To generate *Zfhx4* global knockout mice, heterozygous *Zfhx4* floxed mice were mated with CAG-Cre transgenic mice. Deletion of the floxed allele was confirmed by genomic PCR. Deletion of the Cre transgene was achieved by mating *Zfhx4* heterozygous-deficient mice with C57BL/6 mice as confirmed by genomic PCR. *Zfhx4*-deficient mice were produced by interbreeding of *Zfhx4* heterozygous mice. To

determine mouse genotypes, the following primers were used in PCR: 5′-TCACTGTGCATGAGGCAAAAC-3′ and 5′-GGGTCCCCACTGTGATTTCTA -3′. *Osterix* KO mice were used as described previously[10].

**Microarray analysis**. We analyzed microarray data deposited in the Gene Expression Omnibus (Accession No: GSE126945). Briefly, total RNA was isolated from limb buds of E12.5 ICR mice using an Ambion® WT Expression Kit (Applied Biosystems, Branchburg, NJ, USA). Single-stranded DNA was synthesized from total RNA and subsequently hybridized to a GeneChip®Mouse Gene 1.0 ST Array (Affymetrix, Santa Clara, CA, USA). After 16 h of hybridization, the array was washed, scanned, and finally quantified by an Affymetrix Expression Console (Affymetrix)[10].

**Reverse transcriptase-quantitative polymerase chain reaction (RT-qPCR)**. Total RNA was isolated from cells using a NucleoSpin RNA Plus Kit (Macherey-Nagel, Duren, Germany). Total RNA was denatured by incubation at 65 °C for 5 min, after which cDNA was synthesized using ReverTra Ace Quantitative PCR RT Master Mix with gDNA Remover (Toyobo, Osaka, Japan)[27]. Real-time RT-qPCR amplification was performed using the TaqMan PCR protocol and Step-One plus real-time PCR system (Applied Biosystems, Carlsbad, CA, USA). TaqMan probes used for amplification are listed in Table 1. The expression level of mRNA was normalized to *β-actin* mRNA expression[60].

**Histological and immunohistochemical analyses**. Samples were fixed in 4% paraformaldehyde/phosphate-buffered saline (PFA/PBS) overnight at 4 °C, embedded in paraffin, and cut into 7 μm-thick sections. The sections were deparaffinized, rehydrated, and then stained with Mayer's hematoxylin and eosin (H&E). For antigen retrieval, deparaffinized sections were incubated with 5% hyaluronidase in PBS for 30 min at 37 °C for Col2, Col10, and Mmp13 immunostaining. After blocking with 5% bovine serum albumin (BSA), sections were incubated with anti-Col2 (1:1000; #7050; Woodinville Chondrex, WA, USA), anti-Col10 (1:500, LSL, Tokyo, Japan), anti-Mmp13 (1:200, ab39012; Abcam, Cambridge, UK), anti-Pcna (1:1000; #13110; Cell Signaling Technology, Danvers, MA, USA), or anti-Ki67 (1:200, ab15580; Abcam) antibodies at 4 °C for 16 h, washed thrice with PBS, and then incubated with Alexa Fluor 555-conjugated anti-rabbit immunoglobulin G (IgG) (1:500, A21429; Thermo Fisher Scientific) or anti-mouse IgG (1:500, A21424; Thermo Fisher Scientific)[28,61]. Sample immunoreactivity was analyzed under a fluorescence microscope (Leica DM4 B; Leica Microsystems, Wetzlar, Germany).

**Skeletal preparation**. The skin and internal organs of E15.5, E16.5, and newborn mice were removed and then the mice were fixed in 95% ethanol overnight at room temperature. Cartilage was stained with a 1.5% Alcian blue solution and bone tissues were subsequently stained with a 0.02% alizarin red solution. Skeletal samples were photographed under a stereoscopic microscope (Leica Microsystems)[62].

**In situ hybridization**. Femurs of E12, E13.5, E15.5, and E16.5 mice were fixed in 4% PFA/PBS at 4 °C, washed with PBS, embedded in paraffin, and cut into 7 μm-thick sections. Digoxigenin (DIG)-labeled ssRNA probes were prepared using a DIG RNA Labeling Kit (Roche, Basel, Switzerland). Sections were deparaffinized and treated with 1 mg/ml proteinase K in 0.2 M HCl for 12 min at 37 °C and then subjected to acetylation in 0.1 M triethanolamine/0.25% acetic anhydride. After prehybridization, sections were incubated with DIG-labeled RNA probes at 65 °C for 16 h. The sections were further incubated with anti-DIG-AP Fab fragments (1:2500, Roche) for 2 h. After washing, sections were treated with nitro blue tetrazolium/5-bromo-4-chloro-3-indolyl phosphate (Roche) for various periods. We used a 405 bp fragment of mouse *Col2a1* cDNA, a 637 bp fragment of mouse *Col10a1* cDNA, a 579 bp fragment of mouse *Ihh* cDNA, a 779 bp fragment of mouse *Pthr1* cDNA, a 634 bp fragment of mouse *Runx2* cDNA, an 879 bp fragment of mouse *Osterix* cDNA, a 723 bp fragment of mouse *Zfhx4* cDNA, a 516 bp fragment of mouse *Osr1* cDNA, a 518 bp fragment of mouse *Osr2* cDNA, and a 912 bp fragment of mouse *Pax9* cDNA to generate anti-sense probes. The cDNA probes for *Mmp13*[63], *Sox9*[21], *Ihh*[21], *Ppr*[9], and *Osterix*[10] have been described previously. The cDNA probes for *Col2a1* and *Col10a1* were kindly provided by Noriyuki Tsumaki (Kyoto University, Kyoto, Japan) and used as described previously[26]. cDNA probes for *Zfhx4* (*Zfhx4*-1 mRNA probe; Supplementary information) were synthesized using a DIG RNA Labeling Kit (Roche)[64]. For sections of palatal shelves and limb buds, the *Zfhx4*-2 mRNA probe (Supplementary information) was used.

**von Kossa calcium staining**. Sections were deparaffinized, incubated in a 5% silver nitrate solution for 30 min with exposure to sunlight, and then rinsed with two changes of distilled water. Sections were incubated in a 5% sodium thiosulfate solution for 2 min, rinsed with two changes of distilled water, and then incubated in a nuclear fast red solution for 5 min[10].

**Construction of expression vectors**. Myc-tagged Osterix and Venus-tagged-Osterix expression vectors were used as described previously[10]. The Myc-Runx2 expression vector was generated by subcloning PCR-amplified, full-length *Runx2* cDNA (variant 3, 1791 bp) into the EcoRI and XbaI sites of a pcDNA3 expression vector with six tandem repeats of the Myc tag at the N-terminal. Venus-tagged Runx2 was generated by subcloning the corresponding cDNA into the Venus-tagged expression vector. Flag-tagged Zfhx4 was generated using the Gateway Cloning System (Thermo Fisher Scientific)[65]. The PCR amplification products of Flag-tagged Zfhx4 cDNA were subcloned into the KpnI and XhoI sites of the Gateway entry clone and the resulting clone was transferred into the Gateway pcDNA Zeo destination vector using LR Clonase enzyme mix, thereby constructing the Flag-tagged Zfhx4 expression vector[66].

**Transfection**. Transfection of expression vectors was carried out using a X-treamGene 9 (Roche) in accordance with the manufacturer's protocol.

**Fluorochrome staining and immunocytochemical analysis**. SW1353 cells were transfected with Flag-Zfhx4, Venus-Osterix, or Venus-Runx2 using the X-tremeGENE 9. After 48 h, cells were washed twice with PBS and fixed with 4% buffered paraformaldehyde (WAKO, Osaka, Japan) for 20 min. After treatment with 0.2% Triton X-100 in PBS for 5 min, cells were blocked with 1% BSA in PBS for 1 h, incubated with an anti-Flag (WAKO) antibody at room temperature for 2 h, and then incubated with Alexa Fluor 555-conjugated anti-mouse IgG (1:500,

A21424; Thermo Fisher Scientific) for 30 min. Nuclear staining was performed using Vectashield with 4′,6-diamidino-2-phenylindole (Vector, Burlingame, CA, USA). Samples were visualized using a Leica TCS SP8 confocal microscope (Leica Microsystems).

**Western blot analysis**. Cells were washed with PBS and lysed in lysis buffer [20 mM HEPES, pH 7.4, 150 mM NaCl, 1 mM ethylene glycol-bis[β-aminoethyl ether-N,N,N′,N′-tetraacetic acid, 1.5 mM MgCl$_2$, 10% glycerol, 1% Triton X-100, 10 μg/ml aprotinin, 10 μg/ml leupeptin, 1 mM 4-(2-aminoethyl) benzenesulfonyl fluoride hydrochloride, and 0.2 mM sodium orthovanadate]. Lysates were centrifuged at 12,000 rpm for 20 min at 4 °C and supernatants were boiled in sodium dodecyl sulfate (SDS) sample buffer with 0.5 M β-mercaptoethanol at 95 °C for 5 min. Protein samples were separated by SDS-polyacrylamide gel electrophoresis and then transferred to nitrocellulose membranes. After blocking with 5% BSA in 50 mM Tris-HCl buffer (pH 7.5), the nitrocellulose membranes were incubated with anti-Flag (1:10,000, WAKO), or anti-Myc (1:2000, #ab9132, Abcam) antibodies for 16 h at 4 °C, and then with anti-mouse (1:10,000, Medical and Biological Laboratories) or anti-goat (1:5000, Medical and Biological Laboratories) IgGs conjugated with horseradish peroxidase for 1 h at room temperature. Finally, membranes were visualized using an enhanced chemiluminescence detection reagent (ImmunoStar LD, WAKO)[60].

**Coimmunoprecipitation analysis**. Cells transfected with Flag-Zfhx4, Myc-Osterix, or Myc-Runx2 were washed twice with ice-cold PBS and lysed in lysis buffer. Lysates were centrifuged at 12,000 rpm for 20 min at 4 °C and incubated with an anti-Flag antibody for 16 h at 4 °C, followed by immunoprecipitation with Dynabeads™ Protein G for Immunoprecipitation (Thermo Fisher Scientific). Immunoprecipitates were washed five times with ice-cold PBS and then boiled in SDS sample buffer with 0.5 M β-mercaptoethanol at 95 °C for 5 min. Supernatants were subsequently subjected to western blot analysis[62].

**Statistics and reproducibility**. Randomization and blinding were not performed in the animal studies. Data were statistically analyzed using a two-tailed and unpaired Student's *t*-test for intergroup comparisons. Values of $P < 0.05$ were considered statistically significant. All results were performed three times independently and reproduced with similar results.

**Reporting summary**. Further information on research design is available in the Nature Research Reporting Summary linked to this article.

## Data availability

Microarray data have been deposited in NCBI Gene Expression Omnibus (http://www.ncbi.nlm.nih.gov/geo/) under the accession code GSE126945. The source data for the graphs and charts in the main figures is available as Supplementary Data 1. The original uncropped images referring to Fig. 3a are provided in Supplementary Fig. 13. All other data are available from the corresponding author on reasonable request.

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

## Acknowledgements

We thank Dr. Atsushi Miyawaki (RIKEN, BSI) for the Venus construct. We also thank Edanz (https://jp.edanz.com/ac) for editing a draft of this manuscript. This study was supported in part by Japanese Ministry of Education, Culture, Sports, Science and Technology Grants-in-Aid for Scientific Research (R.N., 16H063930, 16H02686, 25293378, 20K204750, and 21H04841) and "Challenge to Intractable Oral Diseases" (E.N.).

## Author contributions

E.N. and R.N. directed the study. E.N., K.H., and M.A. designed and performed all in vitro and in vivo experiments. E.N., K.H., Y.T., M.A., S.K., and S.Y. performed the molecular and biochemical experiments. T.A. and M.K. generated mutant mice. H.A., N.S., and H.K. generated materials. E.N. assessed the data. E.N., M.A., H.K., T.I., T.Y., T.M., H.A., M.K., and S.N. discussed the data. E.N. and R.N. wrote the manuscript.

## Competing interests

The authors declare no competing interests.
