## [Transparent Peer Review File · Communications Biology]

Reviewers' comments:

Reviewer #1 (Remarks to the Author):

This MS focuses on the function of Zfhx4 in cartilage development. The authors' group revealed that Zfhx4 act together with Runx2 and Osterix for cartilage development via potential physical interaction. Importantly, they generated and analyzed Zfhx4 knockout mice phenotype, showing reduced Mmp13 expression and delayed calcification of cartilage matrices. The experiments were well conducted and the results are clear.

My only concern is the lack of Zfhx4 ISH in limbs.

This information should strengthen the conclusion.

Reviewer #2 (Remarks to the Author):

The authors used expression studies to identify an enrichment of Zfhx4 in cartilage and revealed through global knock-down of the gene that it is required for proper endochondral ossification in mice. The authors further demonstrate a physical and genetic interaction between Zfhx4 and Osterix. The results will likely be of interest to investigators studying the transcriptional regulation of mammalian endochondral ossification. Additional molecular biology experimentation to assess the potential role of Zfhx4 as a transcriptional partner of Osterix as well as additional experimental details would strengthen the authors' conclusions.

Major Comments:

1. As submitted, the Supplemental Figures are too low resolution to interpret. Examples include Supplemental Figure 1b, 1e (skull and palate) and 1f.
2. Additional details should be added to the Results section discussing the floxed allele, including which exons are flanked by loxP sites and how this affects any resulting protein.
3. The authors should list the penetrance for each of the phenotypes they discuss in Zfhx4-mutant embryos and/or postnatal mice. Further, the length of the humerus and femur bones should be measured and compared between wild-type, Zfhx4^{-/-} and Zfhx4^{-/-};Osterix^{+/-} embryos.
4. The results do not conclusively demonstrate that "Zfhx4 functions as a transcriptional partner of Osterix", or that "Zfhx4 can be expected to sequentially interact with Runx2 and Osterix", as stated in the Discussion and Results sections, respectively. Mmp13 promoter assays would address this hypothesis by testing whether adding Zfhx4 enhances Mmp13 expression above expression levels in the presence of Runx2 or Osterix alone. Additionally, the authors could test whether Zfhx4 can physically bind Runx2 in Osterix^{-/-} chondrocytes to assess whether or not a Runx2/Osterix complex is required for Zfhx4 binding. Is the Zfhx4 binding sequence known? Is there a Zfhx4 binding site near Runx2 and/or Osterix binding sites in the Mmp13 promoter? The authors could perform chromatin immunoprecipitation assays to demonstrate that Zfhx4 and Runx2 and/or Osterix bind to the same sites in the Mmp13 promoter.
5. The palatal shelves of Zfhx4-mutant embryos may not elevate because of physical blockage due to the tongue (see Supplemental Figure 4b). The authors could test this possibility by culturing heads with the mandible removed and assaying palatal shelf elevation. The sentence in lines 231-233 should be tempered accordingly. Further, abnormal cell death is a common cause of palatal clefting. The authors should test for this with TUNEL and/or cleaved caspase 3 immunofluorescence analyses.

Minor Comments:

1. OMIM numbers should be added to syndromes discussed in the Introduction and Results sections, such as campomelic dysplasia, Pierre Robin Sequence and 8q21.11 microdeletion syndrome.
2. Additional details should be added to the Methods section pertaining to the Southern blot analyses,

such as the enzyme that was used to digest the DNA and the location of the probe.

3. The levels of expression of *Ihh*, *Ppr*, *Runx2* and *Osterix* do not appear to be different between wild-type and *Zfhx4*-mutant embryos. Given the sentence in the Introduction "Furthermore, *Runx2* also plays a critical role in the regulation of *Col10a1*, *Ihh* and vascular endothelial growth factor", the authors should address why *Ihh* expression levels are not altered in *Zfhx4*-mutant embryos, while expression levels of *Col10a1* (as assessed by in situ hybridization) are dramatically altered. Further, the authors should address why the reduction in *Col10a1* levels could not be confirmed by immunofluorescence and qRT-PCR analyses at the same timepoint.

4. The in situ hybridization probe sequence (or primers used to amplify the probe) for *Zfhx4* should be stated if the probe was used for the first time in this study.

Reply to Reviewer 1:

Reviewer #1 (Remarks to the Author):

This MS focuses on the function of Zfhx4 in cartilage development. The authors' group revealed that Zfhx4 act together with Runx2 and Osterix for cartilage development via potential physical interaction. Importantly, they generated and analyzed Zfhx4 knockout mice phenotype, showing reduced Mmp13 expression and delayed calcification of cartilage matrices. The experiments were well conducted and the results are clear.

We greatly appreciate the reviewer's valuable comment. We are pleased that you found our experiments well conducted and the results clear.

My only concern is the lack of Zfhx4 ISH in limbs. This information should strengthen the conclusion.

We thank the reviewer for this important suggestion. We performed *Zfhx4* ISH using mouse limbs and observed *Zfhx4* expression in the region that surrounded developing fingers, which would be involved in chondrogenesis. We have incorporated the result into Supplementary Figure 1b and described the result in the revised manuscript.

Reply to Reviewer 2:

Reviewer #2 (Remarks to the Author):

The authors used expression studies to identify an enrichment of Zfhx4 in cartilage and revealed through global knock-down of the gene that it is required for proper endochondral ossification in mice. The authors further demonstrate a physical and genetic interaction between Zfhx4 and Osterix. The results will likely be of interest to investigators studying the transcriptional regulation of mammalian endochondral ossification. Additional molecular biology experimentation to assess the potential role of Zfhx4 as a transcriptional partner of Osterix as well as additional experimental details would strengthen the authors' conclusions.

We greatly thank the reviewer for reading our manuscript and providing valuable comments. We are pleased that the reviewer considered our manuscript interesting.

Major Comments:

1. As submitted, the Supplemental Figures are too low resolution to interpret. Examples include Supplemental Figure 1b, 1e (skull and palate) and 1f.

We apologize for the resolution of the Supplementary Figures. We have enlarged the Figures as much as possible. We have revised and enlarged Supplementary Figure 1b and moved the figure to Supplementary Figure 1c. We have enlarged the photographs of the skull and palate in Supplementary Figure 1e and 1f, and moved them to Supplementary Figure 3a and b.

2. Additional details should be added to the Results section discussing the floxed allele, including which exons are flanked by loxP sites and how this affects any resulting protein.

Thank you for the valuable comment. We floxed exons 7 and 8 in *Zfhx4*-floxed mice. Deletion of the exons led to nonsense frame-shift mutation and removed most of the functional domains of the *Zfhx4* protein. We have added this information in the revised manuscript.

3. The authors should list the penetrance for each of the phenotypes they discuss in Zfhx4-mutant embryos and/or postnatal mice. Further, the length of the humerus and femur bones should be measured and compared between wild-type, Zfhx4^{-/-} and Zfhx4^{-/-};Osterix^{+/-} embryos.

We appreciate the valuable comments. We observed 100% penetrance of the cleft palate and dwarf phenotypes in *Zfhx4* KO mice. We have incorporated penetrance of the cleft palate into Supplementary Figure 7a and described the results in the revised manuscript. We have also shown penetrance of the dwarf phenotype below.

To compare wild-type, *Zfhx4*^{-/-}, and *Zfhx4*^{-/-};*Osterix*^{+/-} mice, we mated the mice again, determined the femur length, and assessed the areas of hypertrophic and calcified zones. We clearly confirmed the delayed endochondral ossification in *Zfhx4*^{-/-};*Osterix*^{+/-} mice compared with *Zfhx4*^{-/-} mice as shown below.

Because we were unable to obtain enough wild-type mice in several aged-matched littermates for statistical analyses, we did not incorporate these results into the revised manuscript. If the reviewer believes that it is appropriate to incorporate these data into the revised manuscript, we are willing to include them as Supplementary Figures.

4. The results do not conclusively demonstrate that “Zfhx4 functions as a transcriptional partner of Osterix”, or that “Zfhx4 can be expected to sequentially interact with Runx2 and Osterix”, as stated in the Discussion and Results sections, respectively. Mmp13

promoter assays would address this hypothesis by testing whether adding Zfhx4 enhances Mmp13 expression above expression levels in the presence of Runx2 or Osterix alone. Additionally, the authors could test whether Zfhx4 can physically bind Runx2 in Osterix^{-/-} chondrocytes to assess whether or not a Runx2/Osterix complex is required for Zfhx4 binding. Is the Zfhx4 binding sequence known? Is there a Zfhx4 binding site near Runx2 and/or Osterix binding sites in the Mmp13 promoter? The authors could perform chromatin immunoprecipitation assays to demonstrate that Zfhx4 and Runx2 and/or Osterix bind to the same sites in the Mmp13 promoter.

We appreciate the reviewer's valuable suggestions. First, we performed luciferase reporter assays using three luciferase constructs of the *Mmp13* gene promoter. We found that Zfhx4 itself upregulated *Mmp13* gene promoter activity and increased the effects of Osterix and Runx2 on the *Mmp13* gene promoter. We have incorporated these results into Supplementary Figure 6a. To address the second comment, we attempted to examine the association of Zfhx4 with Runx2 in wild-type and *Osterix* KO cells. However, we did not detect endogenous binding of Zfhx4 to Runx2, although we performed coimmunoprecipitation assays using several antibodies against Zfhx4 or Runx2. We assume that commercially available anti-Zfhx4 and -Runx2 antibodies are unsuitable to detect the endogenous complex because of detection limits and inaccessibility to the protein complex by the antibodies. Therefore, we performed *in vitro* binding assays and immunocytochemistry of chondrocytes and limb bud cells isolated from wild-type and *Osterix* KO littermate mice. Our *in vitro* binding assays showed that Zfhx4 bound to Runx2 in *Osterix*-deficient and wild-type cells. We also found that Zfhx4 and Runx2 showed similar localization patterns in wild-type and *Osterix* KO mouse-derived cells. These results suggest that Osterix is not required for the association of Zfhx4 with Runx2. This is consistent with the notion that Runx2 acts upstream of Osterix (Nakashima *et al.*, Cell 2002; Matsubara *et al.*, J Biol Chem 2008; Nishimura *et al.*, J Biol Chem 2012). We have incorporated these results into Supplementary Figure 5c and d. Third, we performed chromatin immunoprecipitation assays and found that Zfhx4, Osterix, and Runx2 bound to the *Mmp13* gene promoter region. Interestingly, Zfhx4 and Osterix showed similar binding abilities using six primer sets, whereas the association of Runx2 to the *Mmp13* gene promoter was not observed using primers 3–6. Therefore, we speculate that Zfhx4 forms a unique transcriptional complex with Osterix and Runx2. Although we cannot conclude the exact binding sites of Zfhx4, Osterix, and Runx2 on the basis of chromatin immunoprecipitation assays, we assume that Zfhx4, Osterix, and Runx2 directly and indirectly bind to several *Mmp13* gene promoter regions. We have incorporated these results into Supplementary Figure 6b and described them in the

revised manuscript. We believe that the new results strengthen our conclusion. We greatly thank the reviewer for giving us the opportunity to examine this issue in more detail.

5. The palatal shelves of Zfhx4-mutant embryos may not elevate because of physical blockage due to the tongue (see Supplemental Figure 4b). The authors could test this possibility by culturing heads with the mandible removed and assaying palatal shelf elevation. The sentence in lines 231-233 should be tempered accordingly. Further, abnormal cell death is a common cause of palatal clefting. The authors should test for this with TUNEL and/or cleaved caspase 3 immunofluorescence analyses.

We greatly appreciate the reviewer's suggestions. We agree with the reviewer that these points are very important. Accordingly, we have performed organ culture experiments. Surprisingly, we found that palatal shelves derived from *Zfhx4* KO mice were elevated and fused in the organ culture system as newly included in Supplementary Figure 9. Because *Zfhx4* KO mice showed a complete cleft palate at birth (Figures 1 and 4, and Supplementary Figure 7) and we observed *Zfhx4* expression in the palatal shelf, but not in the tongue (Figure 4a), we assume that *Zfhx4* is required to remove the tongue from the middle of developing palatal shelves to generate enough space at the timing of palatal shelf elevation. Therefore, we rewrote and more carefully interpreted these results and discussed this point in the discussion of the revised manuscript. We also rewrote lines 231–233 of the original manuscript. To address the second comment, we performed immunohistochemical analyses using an anti-cleaved caspase 3 antibody in palatal shelves of wild-type and *Zfhx4* KO littermate mice. We did not observe a difference between palatal shelves of wild-type and *Zfhx4* KO littermate mice. Therefore, we consider that a defect in apoptosis was not involved in the etiology of the cleft palate observed in *Zfhx4* KO mice. We have incorporated these results into Supplementary Figure 8 and described them in the revised manuscript.

Minor Comments:

1. OMIM numbers should be added to syndromes discussed in the Introduction and Results sections, such as campomelic dysplasia, Pierre Robin Sequence and 8q21.11 microdeletion syndrome.

Thank the reviewer for your kind suggestion. We have added OMIM numbers for these disorders in the revised manuscript.

2. Additional details should be added to the Methods section pertaining to the Southern blot analyses, such as the enzyme that was used to digest the DNA and the location of the probe.

We thank the reviewer for their important comment. We digested the isolated genomic DNA with EcoRI and performed Southern blotting with a probe for the approximate location described in Supplementary Figure 1c. We have described the information in the revised manuscript. We have also added the probe sequence in Supplementary Information.

*3. The levels of expression of *Ihh*, *Ppr*, *Runx2* and *Osterix* do not appear to be different between wild-type and *Zfhx4*-mutant embryos. Given the sentence in the Introduction “Furthermore, *Runx2* also plays a critical role in the regulation of *Col10a1*, *Ihh* and vascular endothelial growth factor”, the authors should address why *Ihh* expression levels are not altered in *Zfhx4*-mutant embryos, while expression levels of *Col10a1* (as assessed by *in situ* hybridization) are dramatically altered. Further, the authors should address why the reduction in *Col10a1* levels could not be confirmed by immunofluorescence and qRT-PCR analyses at the same timepoint.*

We apologize for the confusion. We agree with the reviewer regarding *Ihh*, *Ppr*, *Runx2*, and *Osterix* expression in wild-type and *Zfhx4* KO mice, and appropriately modified the corresponding sentence in the revised manuscript. Regarding *Ihh* and *Col10a1* expression, several studies have reported that *Runx2* plays a role in regulation of *Ihh* and *Col10a1*. However, we assume that transcriptional regulation of *Ihh* and *Col10a1* is different. Indeed, in our study, *Col10a1*, but not *Ihh*, expression was markedly reduced in *Zfhx4* KO mice (Figure 2c). These results suggest involvement of *Zfhx4* in *Col10a1* expression. However, as the reviewer pointed out, the expression levels of *Col10a1/Col10* were not altered when we determined their expression by immunohistochemical and RT-qPCR analyses (Figure 2d and e), although the delayed expression pattern of *Col10* was consistent with the ISH results. We attempted to perform these experiments at the same time point as much as possible, but a slight time difference in sacrificing mice affected the starting point of *Col10a1* expression, at least in our hands. Therefore, we speculate that *Col10a1* is not a critical transcriptional target for *Zfhx4*. We have more carefully and appropriately described the results in the revised manuscript. We thank the reviewer for reading our original manuscript very carefully.

*4. The *in situ* hybridization probe sequence (or primers used to amplify the probe) for *Zfhx4* should be stated if the probe was used for the first time in this study.*

We apologize for not providing this information. We have added the target sequences of Zfx4 used for ISH and WISH to Supplementary Information. We appreciate the reviewer's suggestion and careful reading of our manuscript.

REVIEWERS' COMMENTS:

Reviewer #1 (Remarks to the Author):

The authors addressed the reviewer's concerns. I, therefore, recommend the acceptance of this manuscript.

Reviewer #2 (Remarks to the Author):

The authors have addressed the majority of my concerns and the experimental additions have improved the manuscript. However, additional clarification should be provided on three previously-raised points.

There are still too few details about the conditional mouse allele that was utilized. What protein domains are encoded by exons 7 and 8? Does the nonsense frameshift mutation lead to nonsense-mediated decay of the entire transcript or, instead, a truncated protein that still retains some function?

Even though only two wild-type mice were analyzed, the authors should incorporate the data relating to the dwarf phenotype and length of the femur, area of hypertrophic zone and area of calcified zone presented in the rebuttal to the manuscript.

Despite statements to the contrary in the rebuttal, the authors have not "more carefully and appropriately described the results in the revised manuscript" regarding Col10. The authors need to raise the point that these experiments were performed at slightly different timepoints and state that they do not believe that Col10a1 is a transcriptional target of Zfhx4.

Reply to Reviewer #1:

The authors addressed the reviewer's concerns. I, therefore, recommend the acceptance of this manuscript.

Thank the reviewer for reviewing our revised manuscript. We are so happy that you have recommended the acceptance of the manuscript.

Reply to Reviewer #2:

The authors have addressed the majority of my concerns and the experimental additions have improved the manuscript. However, additional clarification should be provided on three previously-raised points.

We greatly appreciate your review on our revised manuscript. We are grateful to hear that we have addressed the majority of the reviewer's concerns and the experimental additions have improved the manuscript.

There are still too few details about the conditional mouse allele that was utilized. What protein domains are encoded by exons 7 and 8? Does the nonsense frameshift mutation lead to nonsense-mediated decay of the entire transcript or, instead, a truncated protein that still retains some function?

The removal of the exons 7 and 8, and nonsense frameshift lead to the deletion of all homeobox domains, 13 zinc finger domains and a region coding nuclear localization signal. Therefore, we believe that a truncated protein has no function, even if this is expressed. We incorporated this information in the method section of the revised manuscript.

Even though only two wild-type mice were analyzed, the authors should incorporate the data relating to the dwarf phenotype and length of the femur, area of hypertrophic zone and area of calcified zone presented in the rebuttal to the manuscript.

Thank you so much for a valuable comment. Accordingly, we incorporated the results in the revised manuscript (Supplementary Figure 4 and 6), and described the results in the manuscript.

Despite statements to the contrary in the rebuttal, the authors have not "more carefully and appropriately described the results in the revised manuscript" regarding Col10. The authors need to raise the point that these experiments were performed at slightly different timepoints and state that they do not believe that Col10a1 is a transcriptional target of Zfhx4.

We agree with the reviewer's comment. Accordingly, we have described this point in the discussion of the revised manuscript.